# Prevalence of physical violence against people in insecure migration status: A systematic review and meta-analysis

**Alexandria Innes**[1]*, **Sophie Carlisle**[2], **Hannah Manzur**[3], **Elizabeth Cook**[1], **Jessica Corsi**[4], **Natalia V. Lewis**[5]

1 Violence and Society Centre, School of Policy and Global Affairs, City, University of London, London, United Kingdom, 2 Institute of Psychiatry, Psychology & Neuroscience, King's College London, London, United Kingdom, 3 Violence and Society Centre, City, University of London, London, United Kingdom, 4 Violence and Society Centre and City Law School, City, University of London, London, United Kingdom, 5 Centre for Academic Primary Care, Bristol Medical School, University of Bristol, Bristol, United Kingdom

* alexandria.innes@city.ac.uk

**Data Availability Statement:** All relevant data are within the manuscript and its Supporting information files.

## Abstract

### Objectives

This study summarised evidence on the prevalence of interpersonal, community and state physical violence against people in insecure migration status.

### Methods

We conducted a systematic review and meta-analysis of primary studies that estimated prevalence of physical violence against a population in insecure migration status. We searched Embase, Social Policy and Practice, Political Science Complete, SocINDEX and Web of Science Social Sciences Citation Index for reports published from January 2000 until 31 May 2023. Study quality was assessed using an adapted version of the Joanna Briggs assessment tool for cross-sectional studies. Two reviewers carried out screening, data extraction, quality assessment and analysis. Meta-analysis was conducted in Stata 17, using a random effects model and several exploratory subgroup analyses.

### Results

We retrieved 999 reports and included 31 retrospective cross-sectional studies with 25,997 migrants in insecure status. The prevalence estimate of physical violence was 31.16% (95% CI 25.62–36.70, p < .00). There was no statistically significant difference in the estimates for prevalence of violence for men (35.30%, 95% CI 18.45–52.15, p < .00) and for women (27.78%, 95% CI 21.42–34.15, p < .00). The highest point estimate of prevalence of violence was where insecure status was related to employment (44.40%, 95% CI 18.24–70.57, p < .00), although there were no statistically significant difference in the subgroup analysis. The prevalence of violence for people in undocumented status was not significantly different (29.13%, 95% CI 19.86–38.41, p < .00) than that for refugees and asylum seekers (33.29%, 95% CI 20.99–45.59, p < .00). The prevalence of violence in Asia was 56.01% (95% CI

**Funding:** All authors (AI, SC, HM, EC, JC, NL) were
supported by the UK Prevention Research
Partnership (Violence, Health and Society; MR-
VO49879/1). The funders had no role in study
design, data collection and analysis, decision to
publish, or preparation of the manuscript.

**Competing interests:** The authors have declared
that no competing interests exist.

22.47–89.55, p < .00). Europe had the lowest point prevalence estimate (17.98%, 95% CI
7.36–28.61, p < .00), although the difference was not statistically significant. The prevalence
estimate during the migration journey was 32.93% (95% CI 24.98–40.88, p < .00). Intimate
partner violence attached to insecure status was estimated at 29.10%, (95% CI 8.37–49.84,
p = .01), and state violence at 9.19% (95% CI 6.71–11.68, p < .00).

## Conclusions

The prevalence of physical violence is a concern among people in a range of insecure
migration statuses. Prevalence of violence is not meaningfully higher for people in undocu-
mented status than for people in other types of insecure status.

## Review registration

PROSPERO (CRD42021268772).

## Introduction

Migrants without regular immigration status experience violence disproportionately to the
rest of the population [1]. A technical report assembled by the United Nations Office on Drugs
and Crime in collaboration with Red Cross Red Crescent in 2015 and a report published by
the Council of Europe Committee on Migration, Refugees, and Displaced Persons both evi-
dence violence against migrants [1, 2]. These documents find that there is a lack of accurate
measurement and reporting on irregular or undocumented migration and find that a lack of
regularised status leads to violence for several reasons. These include limited opportunities to
work that push migrants into informal labour markets where they are vulnerable to exploita-
tion [1, 2]; fear to report violence, coercion and exploitation due to the concern that contact
with authorities leads to removal [1]; direct violence against migrants in detention facilities
and in interaction with border guards [1]; and interpersonal violence, xenophobia and hate
crime that targets migrants in the community [1, 2]. The 2015 UNODC report cites evidence
to support the idea that the criminal justice system is not brought sufficiently to bear on
instances of exploitation of vulnerable migrants. There is evidence to support that coupling
immigration control with prosecution of violence leaves violence against migrants unreported
and therefore unprosecuted, due to fear of removal [1, 3]. In such a situation, the state makes
migrants vulnerable by fostering an environment in which migrants cannot access protection
due to fear regarding their own insecure migration status.

   Additional research has identified vulnerability linked to particular forms of migration sta-
tus, including family-based visas that incorporate a 'no access to public funds' stipulation in
the UK [4], or a similar stipulation elsewhere. This uncovers an important set of vulnerabilities
built into immigration statuses that internalise a form of dependency, or that exacerbate an
existing power imbalance. Hence, it is necessary to consider how and when these forms of vul-
nerability are linked to violence. In summary, there are reasons to assume that despite the vari-
ation in forms of insecure migration status, there are common vulnerabilities to violence
shared among them. Measuring the prevalence of violence against people in insecure immigra-
tion statuses can evidence the effects of these vulnerabilities.

### Rationale: Violence against migrants in insecure status

Violence shortens lives, causes harm, and has social and political implications [5]. Violence
constitutes a major risk to public health [6] The effects to physical health include not only

traumatic injury, but also effects on the brain, neuroendocrine system, immune response, cardiovascular disease, premature mortality and mental health conditions such as depression and anxiety [7]. While global movement of people has increased over the last three decades, so too have immigration restrictions in various contexts [8, 9]. Precarity, including and creating vulnerability to violence, has been identified in association with insecure migration status [9–11]. To date there has been no study measuring the prevalence of violence that is commonly experienced across insecure immigration statuses or how insecure migration status intersects with other social determinants of health.

People in insecure immigration status experience physical, psychological, emotional, sexual, verbal, structural and legal violence [12]. This study focuses on physical violence. It was necessary to limit the outcome so that the review was manageable, and measures of physical violence are more widely and consistently available than others [13]. Violence against migrants can be broadly categorised into four different contexts: a) *Violence in transit*. This is usually when a person is travelling from one place to another, usually on a journey to seek a form of protection (for example, political asylum). These journeys are usually characterised by a lack of status, as the forms of transit available are often undocumented and informal [14–21]. b) *Violence in custody*. This includes violence experienced by migrants during arrest—which is often the result only of crossing a border or being present in a country with no status—during detention [22–26], and during removal [27, 28]. This category also includes people in asylum reception centres who have made an asylum application or appealed an asylum decision and are awaiting the outcome of that application or appeal [24]. Due to the often-involuntary nature of asylum reception, people who are housed within asylum reception centres can be considered to be under the custody of the state [22–26]. c) *Work-related violence*. This often emerges as a result of work-visas abroad that are tied to a particular employer, or a particular form of employment. A lack of alternative options, along with a reduced network and often a lack of linguistic and cultural proficiency in the host state produces a vulnerability to abuse and to a form of indentured servitude or modern slavery [29–31]. Literature on human trafficking also deals with modern slavery, indentured servitude, and violence attached to both formal and informal work processes (for examples see [32–35]. d) *Family violence*. This predominantly includes intimate partner violence (IPV) where one or both partners is an immigrant. It includes statuses that have an included dimension of dependency where the immigrant spouse relies on the marital relationship in order to maintain status, such as in the case of spousal visas, exacerbated when they include a financial dependency element such as the UK's *no recourse to public funds* stipulation, or the US's *affidavit of support* requirement [36–38]. Nonetheless, this category can also be expanded to include other forms of domestic violence in mixed status or immigrant families such as child abuse [39, 40] or elder abuse [41–43].

This study aimed to estimate the prevalence of physical violence that is experienced by people in insecure immigration status.

## Materials and methods

We conducted a systematic review and meta-analysis of primary studies that estimated prevalence or allowed a prevalence estimate to be calculated for violence against a population in insecure migration status. This report follows the Cochrane guidance for undertaking a systematic review [44] and the Preferred Reporting Items for Systematic Reviews and Meta-Analyses (PRISMA) reporting checklist [45]. The protocol was prospectively registered on PROSPERO [CRD42021268772] [S1 Appendix].

## Eligibility criteria

We included primary quantitative studies or quantitative components of mixed methods studies of any design if they reported measures of insecure immigration status and physical violence experienced by people of any age while in insecure immigration status. Only peer reviewed reports in English published since 1 January 2000 were included. See Appendix 2 [S2 Appendix] for more detailed information.

## Information sources

Database selection was based on initial scoping, combined with areas of expertise across the authorship. Five databases were selected: *Embase*, *Social Policy and Practice*, *Political Science Complete*, *SocINDEX* and *Web of Science Social Sciences Citation Index*. All selected studies were subject to backwards and forwards citation tracking to identify additional studies for inclusion. Forwards citation tracking was carried out using the tool available in *Google Scholar*. We ran the searches on 22 September 2021 and updated on 31 May 2023, for records from 1 January 2000. The start date was chosen to exclude work that predated immigration reforms in the 1990s.

## Search strategy

We combined three concept clusters 'immigration', 'violence' and 'methods' and employed a Boolean search to link the three concept clusters (AND search) while using multiple descriptive terms in each cluster (OR search) [S3 Appendix].

## Selection process

The first reviewer screened all titles and abstracts against the inclusion and exclusion criteria. Studies that appeared to satisfy the inclusion criteria then underwent full-text screening. Both stages of screening took place in Rayyan. The second reviewer independently screened 20% at both stages. Discrepancies were resolved through discussion and consensus.

## Data collection process

A piloted, bespoke Excel data extraction form detailed sixteen items, which were collected by the first reviewer and then checked for accuracy and completeness by the second reviewer, with discrepancies resolved through discussion and consensus. These included (a) author and year, (b) study design, (c) country of study, (d) source of participants and setting, (e) inclusion criteria, (f) timeframe and type of data collection e.g. retrospective, between date-date, (g) analysis details, (h) sample size, (i) socio-demographics, (j) exposure—how insecure status was measured, (k) timeframe of the exposure, (l) number of participants in insecure status, (m) outcome—how violence was measured, (n) time frame of violence, (o) country of violence, and (p) findings. In cases where data was not specified or disaggregated sufficiently within the study, corresponding authors were contacted to request the raw data or any available disaggregated data, with follow-up requests sent after two weeks. Of five corresponding authors, three responded. Two were able to provide the requested data.

The exposure was insecure status. The conceptualisation of insecure status was generated from a spectrum of statuses including no status, temporary statuses, and dependent statuses. There is no existing formalised definition of insecure migration status. Previous studies have demonstrated that migration is linked with precarity [9], and that power imbalances and dependencies linked to visa status produce specific vulnerabilities to violence [4]. Typically, large population level surveys will record *country of birth*, which means that this is frequently

adopted as the indicator of immigrant status in quantitative research. Nevertheless, there is a difference between immigrant status—a binary category that relies on residing outside one's country of birth—and immigration status, which refers to the category in which one has entered the country and the basis of their permission to remain. We adopted the conceptualisation of *insecure status* to link the characteristic of precarity or vulnerability to a population, allowing the effects of that precarity (rather than the specifics of each separate status) to be measured in a more robust way than is currently possible see S4 Appendix [S4 Appendix, 10].

The outcome was physical violence experienced while in insecure immigration status. The definition of violence that we adopted in this study followed that of the World Health Organization definition and typology of violence [46]. Physical violence in this systematic review included both interpersonal violence and state violence. Interpersonal violence might happen in the home, or in the community. It may be perpetrated by a stranger or an acquaintance but is affected by social relationships at the community level. Collective physical violence was also included in this review, in the context of policies that used physical coercion in their realization, such as forms of immigration enforcement. These policies are used by a collective actor (the state) against a collective that share a determined characteristic (lack of immigration status or in violation of immigration status). The WHO classifies this violence as social, political or economic. Thus, state violence in the context of immigration enforcement that is carried out by as a means of disciplining and removing people who do not meet the criteria for belonging designated by the state can be considered collective political violence. Research has identified coercive policies used at all parts of the immigration processes. This includes things like arbitrary detention; deportation [15, 28, 47, 48]; removal to unsafe locations [49–51]; torture; pushbacks to prevent border crossing even when this leaves people in particularly perilous conditions [52, 53]; maltreatment and abuse when in state custody; sexual violence and rape in state custody; and the use of restraint, assault, and brutality to achieve submission [54, 55]. If an immigration petition is rejected, state authorities might forcibly evict, detain, and remove people, and regularly leaves people without immigration status destitute. In this context the state is adopting policies that often employ physical violence or forms of coercion that result in physical violence; thus, we describe these policies as state violence. It should be noted that the academic literature finds an association between insecure migration status and structural violence as theorised by Galtung [56]. While this is important, it was beyond the scope of this research to locate, identify, and typologize all types of structural violence against people in insecure migration status. That is not to undermine the importance of recognising and addressing the relevancy of structural violence against migrants; time and space constraints required limiting the scope of the project.

## Study risk of bias assessment

We conducted a detailed risk-of bias assessment of all the included studies, using an adaptation of the Joanna Briggs assessment tool for cross-sectional studies [57]. We assessed risk of bias for nine domains: definition of inclusion criteria, description of study subjects and setting, measure of insecure immigration status, identification of confounders, strategies for dealing with confounders, violence measure, reporting of raw data, reporting of association between insecure immigration status and violence. Risk of bias was assessed per study. As per the Joanna Briggs guidance, rather than attribute each study an overall score based on how many domains they met the criteria for, we reported the complete assessment [S4 Appendix]. The risk of bias was assessed independently by two reviewers and any disagreements were discussed, resolved and recorded. Because accurate data on people in insecure migration status is very difficult to obtain, we were cognizant of the difficulty in establishing accurate and

representative data on violence for people in insecure migration status and assessed studies on the criterion of considering risk of bias in pre-existing data records and in interview and survey methods. For example, in studies where violence data were accessed from existing records of a health clinic or women's shelter it is worth considering that there may be significant barriers to access, or barriers to reporting that affect people in insecure status, which has an impact on the accuracy or representativeness of data. In studies where data were collected through interview or survey, we looked for whether they reported the ways in which they considered and accounted for positionality of the researcher and potential bias imposed by the study setting. We also appraised the fitness of the measurement of insecure migration status for the study protocol, the method of measuring violence, and the strategies used to identify and account for confounding factors.

## Synthesis methods

Meta-analysis was conducted using Stata 17 [58]. The raw number of participants experiencing physical violence (the numerator) and the total number of participants in the study population (the denominator) were extracted to calculate prevalence. All studies meeting the inclusion criteria and reporting disaggregated raw data (i.e., the numerator and denominator) or enough information to calculate the numerator and denominator, were included in the synthesis. Where studies reported a percentage and a denominator, the numerator was calculated by one of the reviewers.

A random effects model [59, 60] was used to determine an overall pooled prevalence estimate with 95% confidence intervals (CIs) on physical violence in those with insecure immigration status. Heterogeneity was assessed using the $I^2$ statistic. Forest plots were used to give a visual assessment of the pooled prevalence estimates, 95% CIs and weighting, produced by Stata.

Subgroup analyses [61] were not determined *a-priori*; several post-hoc, exploratory subgroup analyses were carried out, with the aim of exploring possible causes of heterogeneity, and of aiding interpretation of the results. These analyses included by gender as reported in the primary studies, perpetrator of the violence (community versus state versus individual), geographic region, contextual timeframe of violence, and immigration status.

## Results

### Study selection

The database searches produced 14,421 records for screening before de-duplication. After de-duplication, there were 10,652 records. The abstract screening stage identified 1001 full-text reports, from which we included 31 studies [22, 31, 62–91]. All included reports and 23 relevant systematic reviews were tracked backwards and forwards for relevant citations. 27 additional texts were assessed for inclusion, of which 1 was included [Fig 1].

### Characteristics of studies

31 studies with 25,997 participants were included in this review (See Table 1). All were retrospective cross-sectional studies and all studies used non-probability sampling. The majority used convenience or purposive sampling, with the exception of Nakash et al. [71], which used consecutive sampling of a population of asylum arrivals. Of the 31 studies selected, 25 did not report a comparison group for insecure immigration status [22, 63, 64, 66–69, 70, 71, 73–77, 79, 80, 83–91]. Thirteen of the included reports included only female participants [63, 66, 70, 72, 73, 78, 80–82, 86, 87, 92, 93]. One study was exclusively male [76]. Four of the included

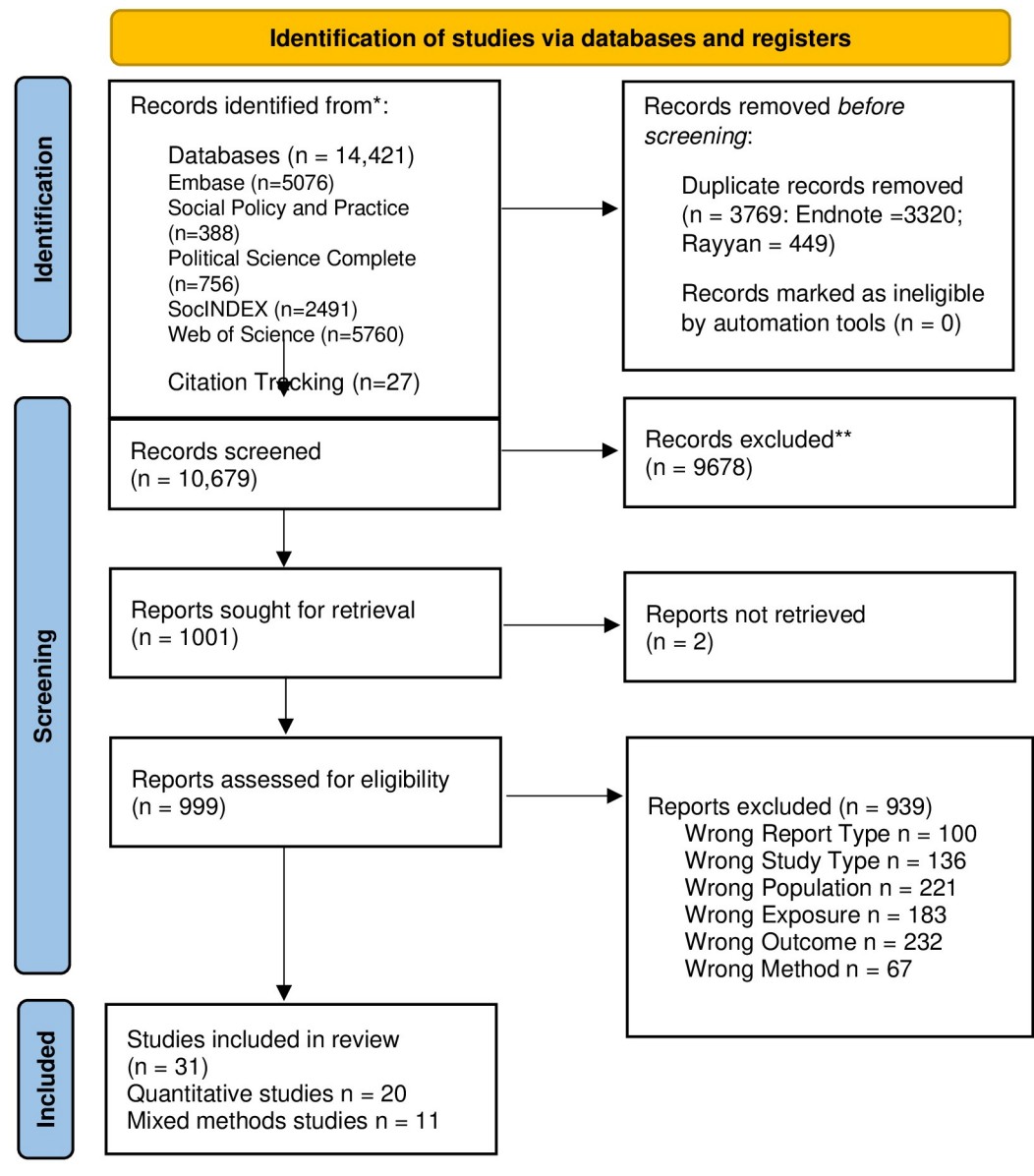

**Fig 1. PRISMA flow diagram.**

studies did include female and male participants although were heavily biased towards male participants [74, 75, 84, 91]. Only three studies reported gender as a non-binary category [69, 85, 91]. While Couture-Carron et al. [92] met the inclusion criteria, because experience of violence was part of the inclusion criteria for that study, and all participants were in insecure status, it could not be included in the prevalence estimation.

Generally, there was a lot of diversity in the way violence was conceptualised and measured. Hadush et al. [86], Logie et al. [88], Ogbonnaya et al. [72], Okenwa-Emegwa et al. [73], Segneri [90], Stewart et al. [78] and Zadnik et al. [82] all adopted a validated tool to measure violence (see Table 1).

**Table 1. Characteristics of included studies.**

| Author (year) | Study design | Country of study | Study Description | Insecure immigration status | N | Gender | Violence type, measure | Measurement tool | Violence timeframe | Country violence |
|---|---|---|---|---|---|---|---|---|---|---|
| Arsenijevic et al. (2017) [22] | Cross sectional | Serbia | Quantitative component of mixed methods study of violence experienced by migrants travelling along the Western Balkan corridor to Northern Europe. Migrants and refugees attending mobile mental health clinics run by Médecins Sans Frontières. | Undocumented | 992 | Mixed, 30% female | Physical trauma caused by acts of violence | Bespoke questionnaire | During journey | Macedonia, Bulgaria, Hungary, Serbia |
| Ben Farhat et al. (2018) [62] | Cross sectional | Greece | Quantitative component of mixed methods study of violence experienced by Syrian refugees in Greece, mental health status and access to information during journey and in Greece. | Refugee / asylum seeker | 728 | Mixed, 41.3% female | At least one violent event | Bespoke questionnaire | During journey | Greece, Turkey |
| Bianchi et al. (2021) [84] | Cross sectional | Italy | Evidence of physical violence and torture in medico-legal reports of asylum seekers in Italy. | Refugee / asylum seeker | 196 | Mixed, 99% male | Blunt instrument beating | Extraction from medical records | During journey | Libya |
| Bouhenia et al. (2017) [83] | Cross-sectional | France | Quantitative study of violence experiences and health in transit towards Calais and in Calais when in informal camp site known as 'The Jungle'. | Refugee / asylum seeker | 402 | Mixed, 95% male | Violence encountered at least once | Bespoke questionnaire | During journey and in Calas | Libya, France, Iran, Sudan, Bulgaria |
| Bronsino et al. (2020) [63] | Cross-sectional | Italy | Quantitative study of sexual gender-based violence experienced by asylum-seeking women during their journey to Europe. Medical records of asylum seekers hosted at the "Teobaldo Fenoglio" Red Cross reception centre in Italy. | Refugee / asylum seeker | 2484 | Female | Sexual / gender-based violence | Extraction from medical records | During journey | Italy Libya |

(*Continued*)

**Table 1.** (Continued)

| Author (year) | Study design | Country of study | Study Description | Insecure immigration status | N | Gender | Violence type, measure | Measurement tool | Violence timeframe | Country violence |
|---|---|---|---|---|---|---|---|---|---|---|
| Coulter et al. (2020) [64] | Cross-sectional | USA | Quantitative study of treatment by the Customs and Border Protection agency experienced by Mexican unaccompanied minors. Face to face surveys in shelters for unaccompanied migrant children in Mexican border towns. | Undocumented | 97 | Mixed, 87% male | Pushed, grabbed, or attacked physically | Bespoke questionnaire | In state custody | USA |
| Dias et al. (2013) [65] | Cross-sectional | Portugal | Quantitative study on prevalence of interpersonal violence among mixed sample of immigrants in Portugal. | Undocumented | 162 | Mixed, 52.4% female | Physical violence | Bespoke questionnaire | Past 12 Months | Portugal |
| Gezie et al. (2019) [66] | Cross-sectional | Ethiopia | Quantitative study locating sexual violence during human trafficking cycle for female Ethiopian returnees. | Undocumented | 671 | Female | Sexual violence (physical) | Bespoke questionnaire | While in trafficking conditions for the 3–24 months preceding study. | Sudan / 'other Arab countries' / South Africa / Europe / Others |
| Gorn et al. (2023) [85] | Cross sectional | Mexico | Descriptive mixed-methods study of migrants transitting through Mexico. Study assessed anxiety symptoms and measured exposure to violence. | Undocumented | 250 | Mixed 53.9% female | Physically attacked, assessed using 'scale on violence during displacement through Mexico.' | Bespoke questionnaire | During journey | Mexico |
| Hadush et al. (2023) [86] | Cross sectional | Ethiopia | Community-based study assessing prevalence of IPV among a random sample of refugee women in the Pinyudo refugee camp. | Refugee/asylum | 406 | Female | Physical violence | Adapted from WHO [46] | Past 12 months | Ethiopia |
| Infante et al. (2012) [67] | Cross sectional | Mexico | Quantitative study of violence against migrants in transit on the Northern Mexican border. Survey conducted by Médecins du Monde, migrants travelling to and returning from the US. | Undocumented | 1512 | Mixed, 90% male | Physical violence | Bespoke questionnaire | While in insecure status | USA |

*(Continued)*

**Table 1.** (Continued)

| Author (year) | Study design | Country of study | Study Description | Insecure immigration status | N | Gender | Violence type, measure | Measurement tool | Violence timeframe | Country violence |
|---|---|---|---|---|---|---|---|---|---|---|
| Islam et al. (2021) [87] | Cross sectional | Bangladesh | Study of violence associated with child marriage among Rohingya refugees in Bangladesh. | Refugee/asylum | 486 | Female | Beating / hitting | Bespoke questionnaire | Past 12 months | Bangladesh |
| Jankovic-Rankovic et al. (2020) [68] | Cross sectional | Serbia | Qualitative study of forced migration experiences, mental well-being and nail cortisol amongst recently settled refugees in Serbia. | Refugee/asylum seeker | 111 | Mixed, 35.1% female | Physical violence | Bespoke questionnaire | During journey | Serbia, journey |
| Leyva-Flores et al. (2019) [69] | Cross-sectional | Mexico | Quantitative study of violence-experiences of migrants in transit through Mexico to the US. Data gathered at five 'Casas del migrante' at strategic points along migrant transit route. | Undocumented | 12023 | Mixed, 77.72% male, 21.73% female, 0.56% trans | Overall violence (includes kidnapping, theft, beating and rape) | Bespoke questionnaire | ??? | Mexico |
| Logie et al. (2022) [88] | Cross sectional | Uganda | Study of substance use, violence, HIV and AIDS among refugee youth | Refugee/asylum | 329 | Mixed 74.8% female | Physical abuse | Brief Inpatient Screen for Intimate Partner Violence [95] | Past 12 months | Uganda |
| Meyer et al. (2019) [31] | Cross sectional | Thailand | Quantitative study of gender differences in abuse among migrant workers on the Thailand-Myanmar border | Employment-related | 589 | Male | Physical abuse | | During journey | Thailand, Myanmar |
| Morof et al. (2014) [70] | Cross-sectional | Uganda | Gender-based violence and mental health among female urban refugees and asylum seekers in Kampala, Uganda. | Refugee / asylum seeker | 117 | Female | Physical violence | Bespoke questionnaire | While in insecure status | Uganda |
| Nakash et al. (2015) [71] | Cross sectional | Israel | Exposure to traumatic experiences among asylum seekers from Eritrea and Sudan during their migration to Israel. | Refugee / asylum seeker | 1044 | Male | Physical violence | Bespoke questionnaire | During journey | Egypt |

*(Continued)*

**Table 1.** (Continued)

| Author (year) | Study design | Country of study | Study Description | Insecure immigration status | N | Gender | Violence type, measure | Measurement tool | Violence timeframe | Country violence |
|---|---|---|---|---|---|---|---|---|---|---|
| Ogbonnaya et al. (2015) [72] | Cross sectional | USA | Association between domestic violence and immigration status among Latina mothers in the child welfare system. Data from National Survey of Child and Adolescent Well-being. Parent is unit of analysis. | Undocumented | 77 | Female | Domestic violence | Conflict Tactics Scale 2 [96] | Last 12 months. | USA |
| Okenwa-Emegwa et al. (2021) [73] | Cross sectional | Sweden | Exposure to violence among Syrian refugee women pre-flight and during flight. Questionnaires and databases coordinated by Statistics Sweden. | Refugee / asylum seeker | 452 | Female | Physical violence | Refugee Trauma History Checklist [97, 98] | During journey | Journey |
| Phillips et al. (2006) [74] | Cross sectional | El Salvador | Treatment of deportees during arrest and detention. | Undocumented / insecure status | 300 | Mixed, 95% male | Physical force used during detention | Bespoke questionnaire | In State Custody | USA |
| Phillips et al. (2002) [75] | Cross sectional | EL Salvador | Use of force in the arrest of immigrants in the US. | Undocumented / insecure status | 211 | 92% male | Physical force used during detention | Bespoke questionnaire | In State Custody | USA |
| Pocock et al. (2018) [76] | Cross sectional | Thailand, Cambodia | Mixed methods survey of migrant and trafficked fishermen in the Mekong. Quantitative survey data from structured interviews with male survivors of trafficking for commercial fishing, in the care of post-trafficking services | Employment-related | 275 | Male | Violence (less severe and more severe combined) | Bespoke questionnaire | During overseas employment | China, Myanmar, Laos PDR, Thailand, Cambodia, Vietnam (Greater Mekong Subregion). |
| Reques et al. (2020) [77] | l Cross sectional | France | Violence experienced by migrants transiting through Libya. Data from migrants consulting the Médecins du Monde reception and healthcare centre in Seine-Saint-Denis. | Undocumented | 72 | Mixed, 23.6% female | Episodes of physical violence | Bespoke questionnaire | During Journey | Libya |

(*Continued*)

**Table 1.** (Continued)

| Author (year) | Study design | Country of study | Study Description | Insecure immigration status | N | Gender | Violence type, measure | Measurement tool | Violence timeframe | Country violence |
|---|---|---|---|---|---|---|---|---|---|---|
| Segneri et al. (2022) [90] | Cross sectional | Italy | Study on evidence of violence in medico-legal assessments inside a first aid and reception centre on Lampedusa. | During journey | 112 | Mixed, 73.2% male | Physical violence and torture | Istanbul Protocol [99] | During Journey | Egypt, Libya, Tunisia, Niger, Chad, Senegal, Mali, Burkina Faso, Cote Ivoire, Algeria, Dubai, Jordan, Turkey, Israel, Rwanda, Uganda |
| Scott (2022) [91] | Cross sectional | Sweden | Study of how young people who sought refuge in Sweden negotiate access to protection. | Refugee / asylum | 85 | Mixed, 94% male | Physical violence | Bespoke questionnaire | Not specified | Sweden |
| Stewart et al. (2012) [78] | Cross sectional | Canada | Health of recent migrant women who experienced violence associated with pregnancy. Data from Childbearing Health and Related Service Needs of Newcomers database. | Mixed | 1025 | Female | Physical abuse associated with pregnancy | Abuse Assessment Screen [100] | Past 12 Months | Canada |
| Suyanto et al. (2020) [79] | Cross sectional | Indonesia | Descriptive study of the lives of Indonesian illegal migrant workers. | Employment-associated | 400 | Mixed, % by gender not reported | Violent treatment—beaten (rarely, often, always combined) | Bespoke questionnaire | During overseas employment. | Hong Kong, Malaysia, Taiwan, Saudi Arabia, other. |
| Vila and Pomeroy (2020) [80] | Cross-sectional | USA | Effects of violence on trauma among immigrant women from Central America in USA. | Undocumented | 108 | Female | Victim of violence (robbery, assaults, abuse, discrimination, extortion, threats) | Bespoke questionnaire | During Journey | Mexico, USA |
| Vives-Cases et al. (2014) [81] | Cross-sectional | Spain | Social and immigration factors in intimate partner violence among Ecuadorians, Moroccans and Romanians in Spain. Fixed quota of 535 participants per country of origin and residential area. | Undocumented | 30 | Female | Current physical intimate partner violence | Bespoke questionnaire | Past 12 Months | Spain |
| Zadnik et al. (2016) [82] | Cross sectional | USA | Effects of undocumented status on rates of victimization and help-seeking among Latinas. | Undocumented | 91 | Female | Physical victimization | Lifetime Trauma and Victimization History Tool [101] | While in insecure status | USA |

We collected data on violence occurring in 44 different countries, grouped by region: Europe [22, 62, 65, 68, 81, 91]; North America [64, 67, 69, 72, 74, 75, 78, 80, 82, 85]; Asia [31, 76, 87]; and Africa [70, 71, 77, 84, 86, 88]. Twenty-five of the included reports measured violence in a single country [64, 65, 67, 69, 70–72, 74, 75, 77, 78, 81, 82, 84–89, 91, 93, 94]. The remaining reports either specified a combination of locations in which the violence occurred or specified that the violence happened while in transit during the migration journey.

Studies measured insecure immigration status in a number of different ways. These included 'undocumented' [22, 64–67, 69, 72, 74, 75, 77, 80–82, 85, 89], 'refugee /asylum seeker' [62, 63, 68, 70, 71, 73, 83, 84, 86–88, 90, 91], 'spousal /family visa', 'employment related' [31, 76, 79]. Two studies measured more than one status but sufficiently disaggregated the data to allow for insecure status to be verified [78].

## Risk of bias in studies

The majority of studies adequately defined their inclusion criteria (55%) and described their study subjects and setting (84%). Whilst some measured insecure immigration status using objective and standard criteria, the way other studies measured insecure immigration status was unclear (52%), causing us to question the validity for the purposes of our study. Most studies failed to identify and account for confounding variables (71%) and most provided insufficient or unclear information on outcome measurement for the purposes of our study (52%) [S3 Appendix].

## Results of individual studies

The overall estimate of prevalence of physical violence for people in insecure migration status was 31.16% (95% CI 25.62–36.70, p < .00) [Fig 2]. However, there were high levels of heterogeneity ($I^2$ = 99.70%) which makes conclusive inferences about the prevalence of violence for people in insecure migration status difficult. Subgroup analyses were conducted to explore the heterogeneity.

## Subgroup analyses

We grouped the results to test our hypothesized expectations. We looked for prevalence of violence against people in insecure migration status by gender as a social determinant of health, and also by status type, by region, and by the time-frame in which violence occurred. Status type, region and time-frame all intersect with other social determinants of health that are not measured in this study. In each of these categories there was too much heterogeneity in the data to offer robust prevalence estimates from pooled data.

**Gender.**   Most of the studies reported exclusively binary gender categories, or reported only on a single gender. Only three studies reported non-binary gender categories [69, 85, 91]. The prevalence of physical violence estimate for men (35.30%, 95% CI 18.45–52.15, p < .00) was not significantly different from the estimate for women (27.78%, 95% CI 21.42–34.15, p < .00) [Fig 3], although the confidence intervals overlap.

In total, eleven studies [63, 66, 70, 72, 73, 78, 80–82, 86, 87] were all exclusively interested in violence against women. Of these eleven studies, five were not specifically about IPV [63, 66, 70, 73, 80]. Stewart et al. [78] deals with violence associated with pregnancy, which includes but is not limited to IPV. Within the studies on IPV, it is likely that violence is underreported by people in insecure status because of the potential threat reporting poses to status.

**Immigration status type.**   The subgroup analysis by immigration status type was driven by the data available. We grouped research as follows: studies in which the population of interest were all in undocumented status at the time of violence; studies in which the population of

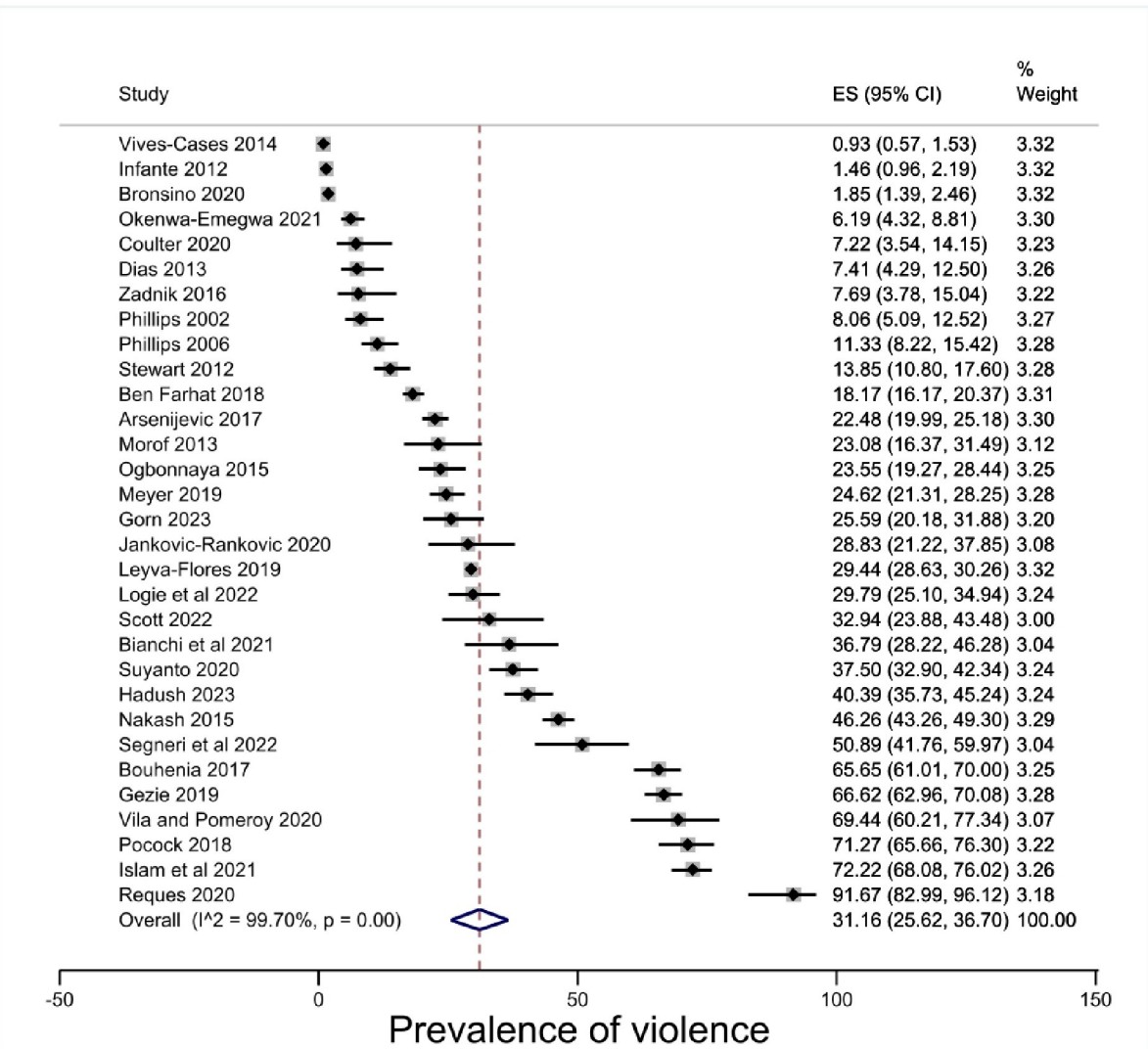

**Fig 2. Prevalence of violence against people in insecure migration status.**

interest were all in refugee or asylum seeker status at the time of violence (and therefore imply violence pre-migration as a potential confounding factor); those in a status related to their employment; and those where the status groupings are mixed or otherwise unclear while still meeting the exposure of insecure status as an inclusion criterion. The estimate of prevalence where insecure status was related to employment was 44.40% (95% CI 18.24–70.57, p < .00). The estimate of prevalence of violence experienced by people in undocumented status was 29.13% (95% CI 19.86–38.41, p < .00) and violence experienced by refugees and asylum seekers was estimated at 33.29% (95% CI 20.99–45.59, p < .00) [Fig 4]. The confidence intervals overlapped and there was no statistically significant difference between the estimates.

**Geographic region.**   The subgroup of geographic region was also too heterogenous to offer a robust measure of prevalence. Indeed, the findings demonstrate diversity and complexity within geographic regions, although no sample can be considered fully representative of the

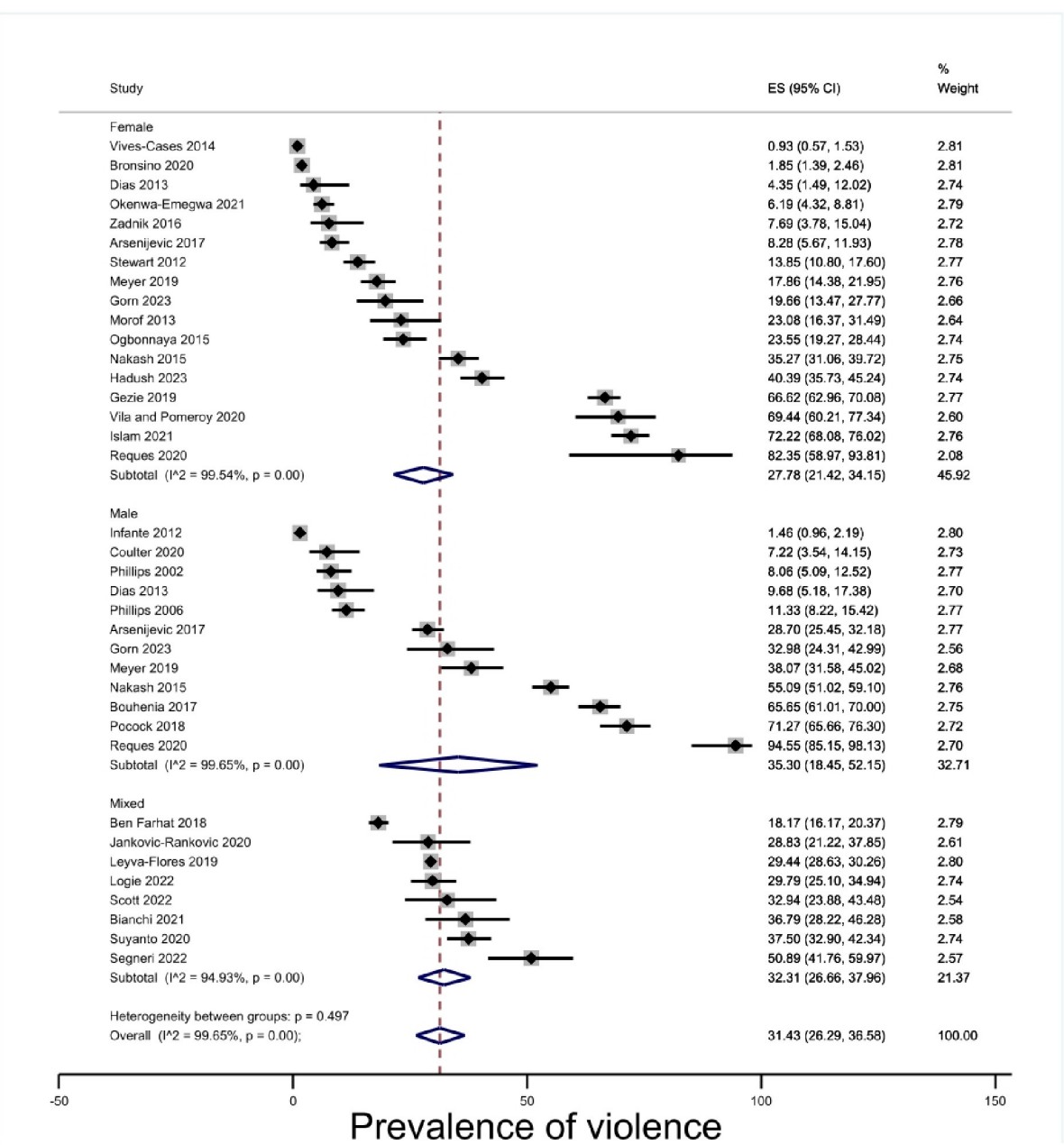

**Fig 3. Prevalence of violence by gender.**

regional grouping. The estimate of prevalence of violence in Asia was 56.01% (95% CI 22.47–89.55, p < .00). The three included studies involved Bangladesh [87], and several East Asian countries included in two studies [31, 76]. The sample is limited even compared to the other geographic regions. Europe 17.98% (95% CI 7.36–28.61, p < .00) and North America 19.53% (95% CI 8.30–30.77, p < .00) had lower point estimates [Fig 5], but confidence intervals were overlapping and there was no statistically significant difference.

**Timing of violence.** Grouping studies by time-frame was guided by the timing of the violence recorded in the included studies. For example, migration journeys might vary in their

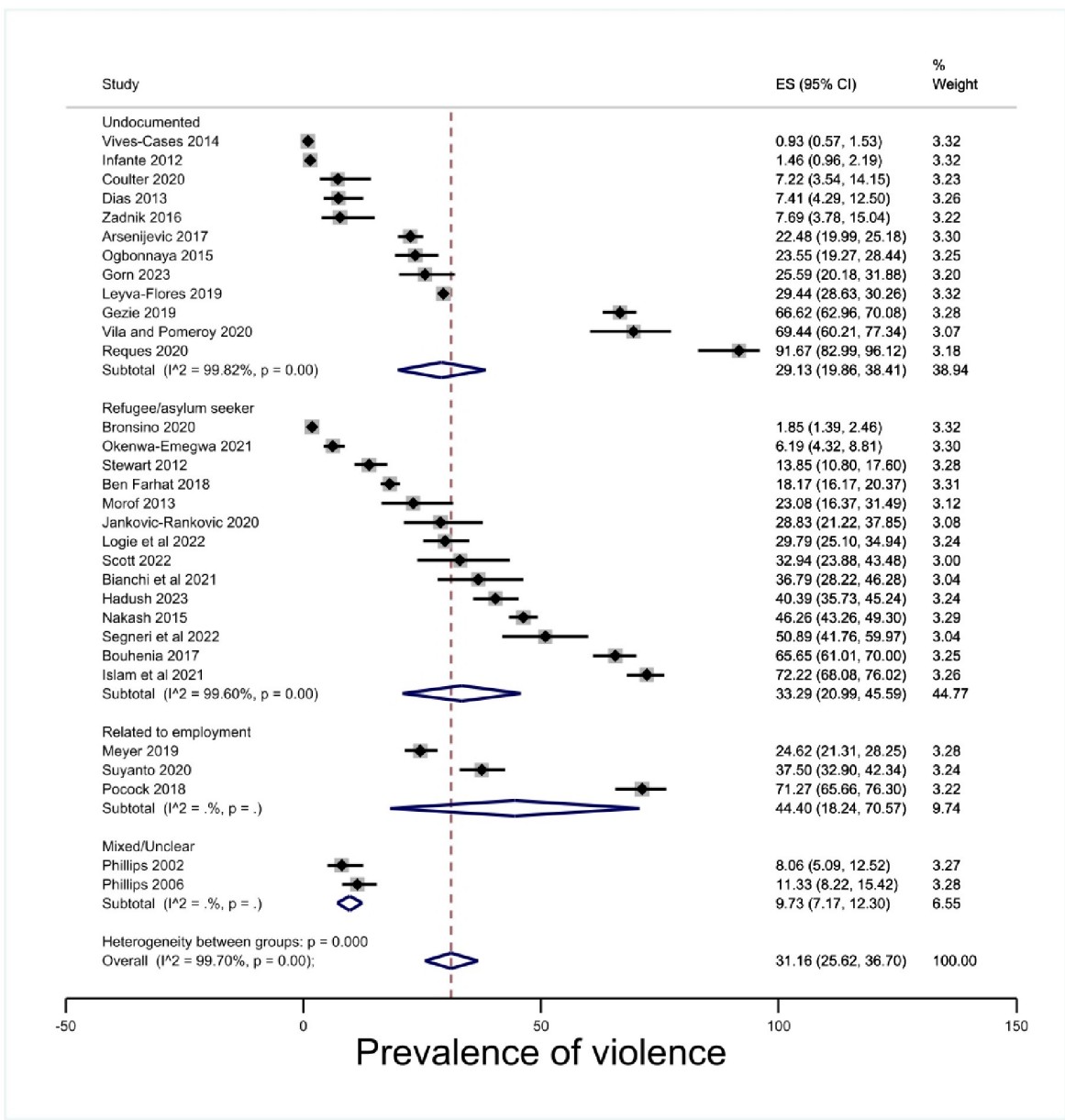

**Fig 4. Prevalence of violence by status type.**

duration, but represent a time of particular vulnerability. Similarly, the amount of time a person spends in state custody might vary, but as a period is defined by the characteristic of being incarcerated. This grouping also included 'past 12 months' to capture studies that shared this measurement characteristic but did not fall into one of the other categories; for example, Dias et al. [65] measured community violence, while Hadush et al., Islam et al., Logie et al., Ogbonnaya et al. and Vives-Cases et al. [72, 81, 86–88] all measured domestic violence. The 'not specified' category grouped all remaining studies where violence happened after arrival in the receiving country but did not include the time period or one of the aforementioned

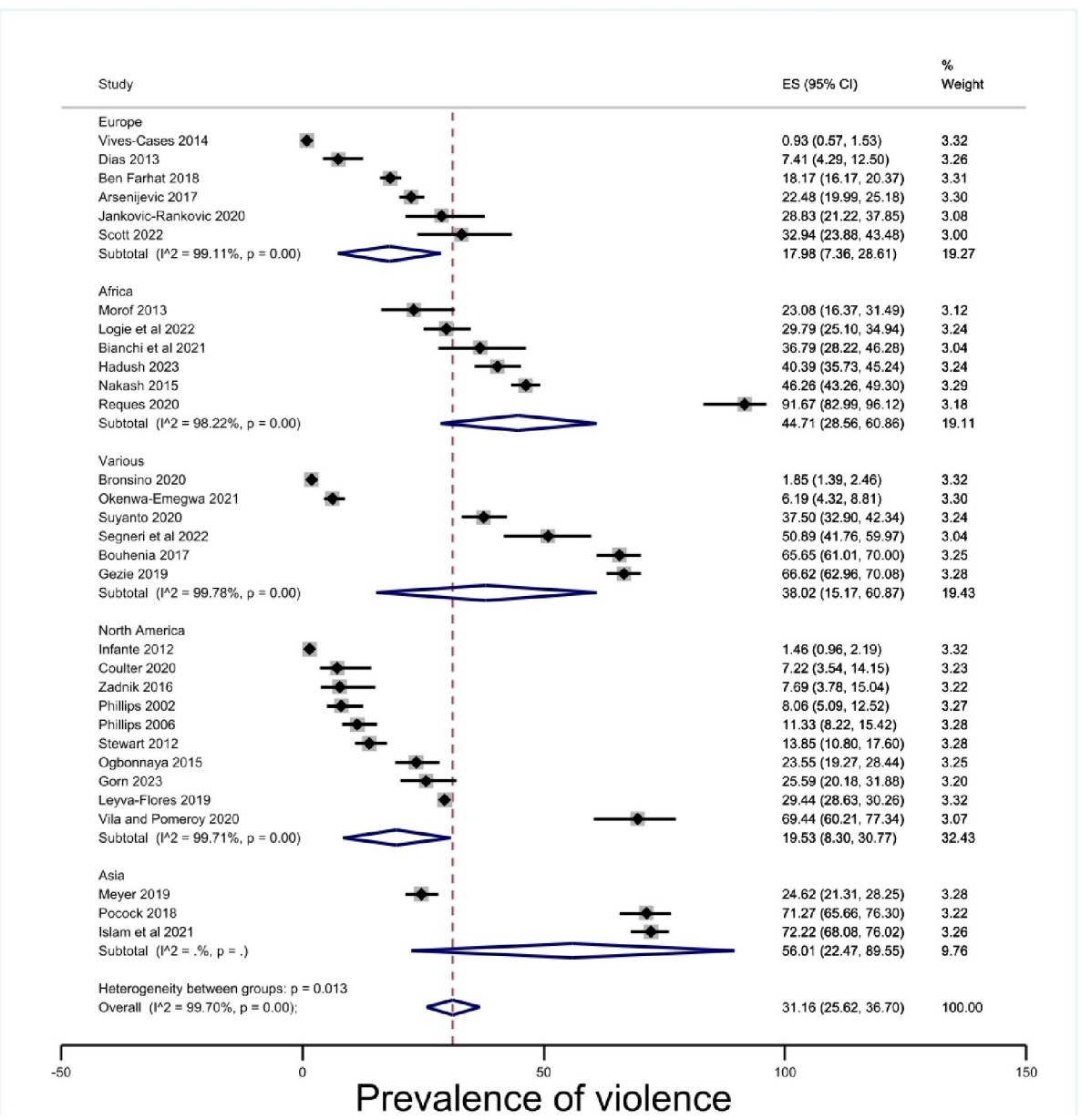

**Fig 5. Prevalence of violence by region.**

characteristics. The prevalence during the migration journey was estimated at 32.93% (95% CI 24.98–40.88, p < .00) [Fig 6].

**Perpetrator.** The perpetrator groupings separated studies that specified state violence and IPV into defined categories. Prevalence of intimate partner violence attached to insecure status was 29.10% (95% CI 8.37–49.84, p < .00) and the estimate for state violence was 9.19% (95%CI 6.71–11.68, p < .00), but the data was particularly limited in the state violence category with only three included studies [64, 74, 75], and remained too heterogenous for a robust estimate [Fig 7]. As with other estimates, it should be noted that the confidence intervals overlapped.

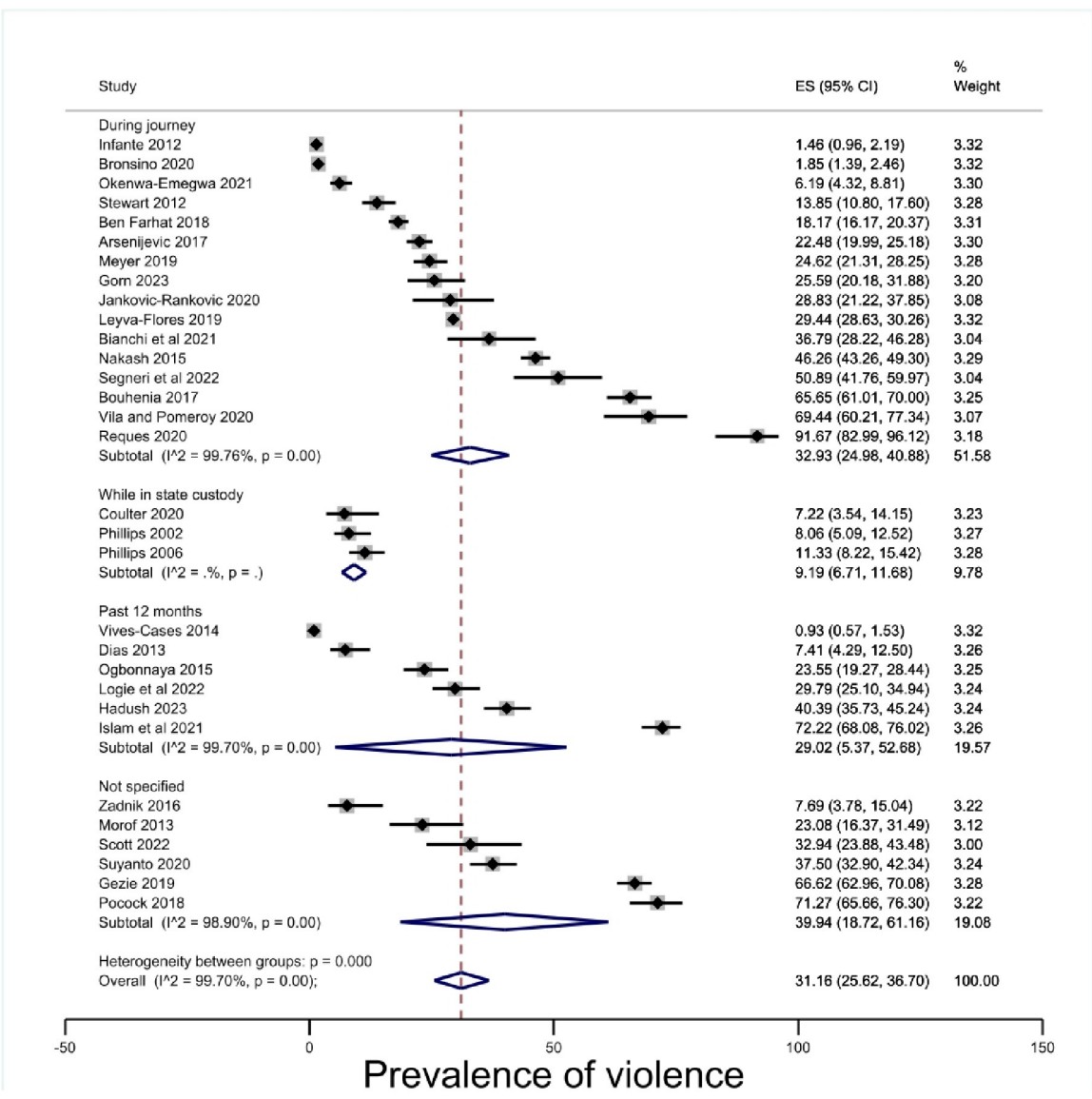

**Fig 6. Prevalence of violence by timeframe.**

## Discussion

### Summary of main results

This systematic review and meta-analysis of 35 cross-sectional studies with a total of 26,116 migrants in insecure status found a prevalence estimate of physical violence 30.86% (95% CI 25.40–36.31, p < .00). When disaggregated by gender prevalence of physical violence was estimated as 35.30% (95% CI 18.45–52.15, p < .00) for men and 27.78% (95% CI 21.42–34.15, p < .00) for women. When disaggregated by insecure status type, prevalence of physical violence was estimated at 44.40% (95% CI 18.24–70.57, p) for employment-based migration, 33.29 (95% CI 20.99–45.59. p < .00) for refugee and asylum seeker statuses, and 29.13% (95% CI 19.86–38.41, p < .00) for undocumented statuses. When disaggregated by geographic region,

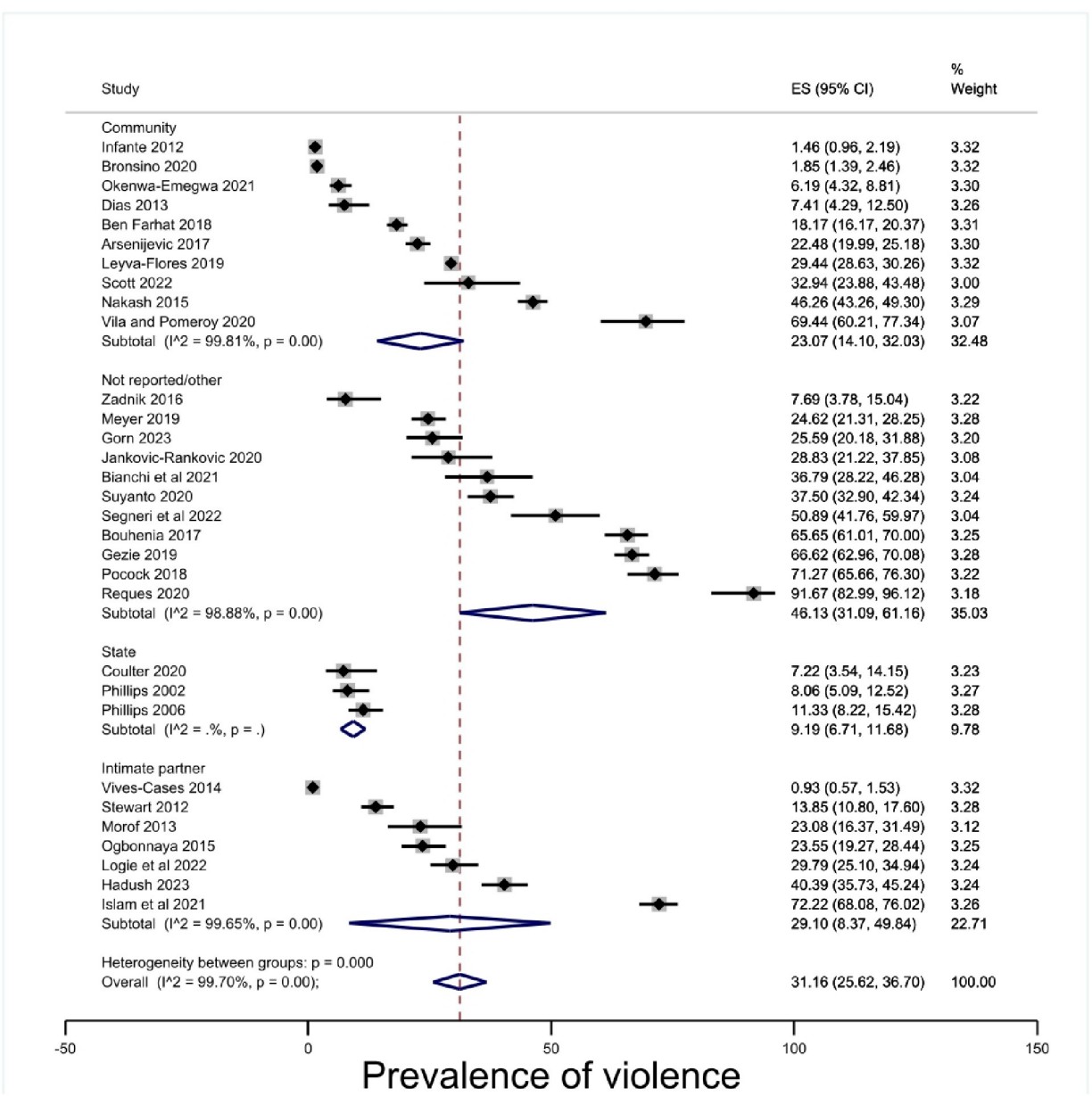

**Fig 7. Prevalence of violence by perpetrator.**

prevalence of physical violence was estimated at 17.98% (95% CI 7.36–28.61, p < .00) in Europe, 19.53, 95% (CI 8.30–30.77, p < .00) in North America, 56.01, (95% CI 22.47–89.55, p < .00) in Asia, and 44.71% (95% CI 28.56–60.86, p < .00) in Africa. When disaggregated by the time during which the physical violence occurred, prevalence was estimated at 32.93% (95% CI 24.98–40.88, p < .00) during the migration journey, 9.19% (95% CI 6.71–11.68, p < .00) while in state custody, and 29.02% (95% CI 5.37–52.68, p < .00) during the 12 months previous to the study. When disaggregated by perpetrator, prevalence was estimated at 9.19% (95% CI 6.71–11.68, p) perpetrated by the state, 29.10 (95% CI 8.37–49.84, p < .01) perpetrated

by an intimate partner, and 23.07% (95% CI 14.10–32.03, p < .00) perpetrated in the community.

Our analysis was informed by the social-ecological model of violence [102] and limited by the data on social determinants of violence reported in the included studies. The social determinants of violence that we discussed include gender and immigration status because these two risk factors were consistently reported across the studies. Other social determinants of violence (e.g., age, socio-economic conditions, social norms, laws/policy/institutions, health) might have contributed to the varying estimates, but data on these determinants was not available across studies. There was no consistency across studies in terms of variables reported, so we were unable to extract and include in our study.

While the confidence intervals for prevalence of violence r for women (27.78%, 95% CI 21.42–34.15, p < .00) and for men (35.30%, 95% CI 18.45–52.15, p < .00) in insecure status overlapped, it is important to note that this measure of physical violence does not capture structural or systemic violence. Intersectional characteristics such as socioeconomic status, ethnicity, education and community context which have a bearing on the measurement of gendered violence were not available for this study. Moreover, a scoping review [103] found that studies of gender-based discrimination and violence were experiential and focused on perceptions and opinions. This type of data was not captured within this systematic review. It is plausible to assume that prevalence of violence against women in insecure status is underestimated.

Women on spousal visas are subjects of the broader vulnerabilities that are connected with (insecure) immigrant status, such as avoiding surveillance and reporting, and with vulnerabilities connected to other co-occurring identity characteristics, such as patriarchal, racist, and gendered social structures [104, 105]. For example, Morash et al. [93] point to the important gendered disparities in status that contribute to the likelihood of abuse if immigration is sponsored by an intimate partner and particularly if the woman immigrated as a 'picture bride' (that is, they were selected from marriage based on a photograph rather than the development of a relationship or an in-person meeting). This indicates that a woman is being selected based on appearance or other known factors rather than her personhood, suggesting objectification. It is also worth noting that Ogbonnaya et al. [72] find no difference in the data for substantiated cases of domestic violence between Latina women with citizenship or legal residence and those who are in unauthorised status in the USA. They theorise that this indicates underreporting on the part of those in unauthorised status because of evidence that there are alleged higher rates that remain unsubstantiated (unreported), and which cannot be explained by other cultural factors because immigrant women who are legal residents provide a control group for cultural factors [72].

This review suggests that physical violence is a widespread issue for people in insecure immigration statuses. The prevalence estimate for violence perpetrated by an intimate partner against people in insecure immigration status was 29.10 (95% CI 8.37–49.84, p < .01) and higher than the reported prevalence estimate for physical IPV in the general population (estimated at 23.1% for women in industrialised English-speaking countries [106]). It is worth noting that studies on IPV (which, in this sample of quantitative research are limited to the Bangladesh, Canada, Ethiopia, Spain, Uganda and the USA) find that there is a vulnerability to violence that can be connected to the vulnerability inherent in the dependent immigration status, and that this is an intersectional vulnerability, linked to gender and other social determinants of health such as ethnicity and community factors.

The prevalence estimate of physical violence associated with employment-based immigration statuses is high (44.40, 95% CI 18.24–70.57), yet these studies were located only in Southeast Asia among specified populations. Meyer et al. [31] estimated prevalence of physical

violence among migrant workers at the Thai-Myanmar border at 24.62% (95% CI 21.31–28.25). Comparable data in Thailand data is only available for specific sectors; for example, workplace violence among nurses was estimated at 12.1% [107]. Suyanto et al.'s [79] study of Indonesian migrant workers estimated prevalence of physical violence at 37.50% (95% CI 32.90–42.34). A survey led by the International Labour Organization and the 'Never Okay' Project found that in Indonesia 70.93% of 1173 survey respondents has experienced violence and harassment at work. Where immigration is connected to employment, it is likely that failure to disclose violence is high due to fear of losing employment and immigration status, thus there is reason to believe even the figure of 44.40 is underestimated.

Physical violence relating to specific subgroups was subject to the same problems as the overall prevalence estimate. Overall, the research that looks at IPV in the context of insecure immigration status raised several points of vulnerability, which include increased likelihood of abuse based on immigration-related factors. These include increased stress levels, lack of community support, social isolation of victim [78, 81, 108, 109] and power disparity embedded in family-based visas [104, 105], and reduced likelihood of reporting to either victim-supporting organisations, migrant-supporting organisations, or the police [72, 93]. The reduced likelihood of reporting might be based on fear of losing status but also on other factors such as lack of knowledge of how and where to report, lack of understanding of implications of reporting, language difficulties, and social isolation. These issues are not isolated to spousal visas, but potentially affect all types of insecure statuses.

Prevalence of violence was estimated in the subgroup of legal status by the subcategories of 'Undocumented', refugee/asylum seeker, and 'employment-related'. There was no sub-category for spouse or family-dependent because this data was not available as a disaggregated category in the included studies. While data within the groupings according to legal status was still too heterogenous to offer any pooled measure of prevalence, it is worth noting that this indicates that there is no meaningful pattern of violence attached to undocumented status when compared with other categories of insecure status. Similarly, it is not possible to draw a link from this data between violence pre-migration (associated with refugee and asylum statuses) and violence post migration. This suggests that it is worth investigating further what status trends and types of insecurity can be associated with high prevalence of violence. It is clear that insecure status produces vulnerability to violence, and that vulnerability is not limited only to people in undocumented status. People in other types of documented and regular immigration status that embeds a form of insecurity are vulnerable to violence.

The regional groupings again were too heterogenous to provide prevalence estimation. Furthermore, too few countries were represented in each category to say anything meaningful about the separate regions. Nevertheless, what is clear from these groupings is that there is a deficit of quantitative data on insecure migration in South America and in Asia (although this study reports only English language sources, which is a source of bias). More research is available focusing on Europe and North America, which is likely driven by data availability and research funding. Thus, we do not have a clear picture of the prevalence of violence for people in insecure migration status globally.

**Strengths and limitations of the review.** The review protocol was pre-registered in the publicly available database to ensure transparency. Two reviewers were involved in every step of the study. We used a comprehensive search strategy across five electronic databases. By not searching other databases we might have missed some studies, although we carried out reference and citations chaining to mitigate this possibility. The search was limited to academic peer-reviewed studies; we did not include grey literature such as reports published by international institutions or third sector organisations. The limited definition of physical violence meant that we excluded studies that did not disaggregate forms of sexual violence into physical

and verbal or coercive control. We also did not include studies of human trafficking unless they specified events of physical violence. Even adopting a limited definition of physical violence, there was too much heterogeneity to allow for robust pooled prevalence estimates. While there is a risk of publication bias influencing prevalence measures, we did not include funnel plots to assess publication bias because they tend to give erroneous results when pooling prevalence data [110, 111].

All included studies used retrospective cross-sectional design and had methodological limitations. Most studies did not control for the core confounders for immigration status and violence. Unmeasured confounders could result in biased prevalence estimates. All studies relied on retrospective recall of the exposure and outcome which is likely to lead to either an underestimate or overestimate of the prevalence of violence. Self-report of violence is likely to have resulted in under-reporting due to the vulnerabilities inherent in insecure migration status, such as fear of surveillance or removal [72, 93]. While several studies used or partially used a standardised tool for the measurement of violence, most did not. One study [84] reported several types of violence, of which we used only the most frequently occurring category to avoid double counting. However, this introduced a risk of underreporting violence in that study.

Because a systematised definition of insecure immigration status does not exist (across borders or across academic studies), our expectation of high levels of heterogeneity across the data was well-founded. The extent to which this data is partial, fragmented and unsystematised is clear in this review. Nevertheless, we can assess the theorised sub-groups and make recommendations for future research. Our study summarised the evidence of violence that is available and highlighted the deficit of standardization across studies relating to conceptualising and measuring insecure migration status. This contributes to the difficulty of estimating prevalence of violence against people who share this particular exposure.

**Implications for policy, practice and future research.** This systematic review can make several recommendations for future research. These include conceptualising a means of measuring the insecurity that is attached to immigration status. This should be differentiated from the state of being an immigrant, which can be measured simply by being 'foreign born'. Being foreign born does not capture the experience of immigration status because people might have more than one citizenship, or may naturalise, or access a secure permanent resident status. While there are of course things that can be broadly or probabilistically attached to being foreign born, this does not articulate the role of immigration status in experiences, and in the case of this review in experiences of violence. A plethora of different immigration statuses with various levels of inherent insecurity have emerged since the 1990s, as have the penalties attached to being without status. This shift is not generally reflected in quantitative data and measurements, leaving huge gaps in what is known about the experience of immigrating. A means of conceptualising and measuring insecure migration status would allow for data to be pooled more easily and therefore would allow measurement of various experiences unique to immigration statuses, which could be useful in many fields, including but not limited to health and social care, crime and policing, demography, and politics. Nevertheless, it should be recognised that requiring data on insecure migration status can further deter migrants from engaging with the state or with service providers in any way in case it compromises their security. And the fear of engagement compromising security is well-validated. For example, the hostile environment in the UK has implemented extensive data-gathering and surveillance practices with the intention of deterring undocumented or irregular immigration, and of removing those who are in an irregular or invalid status [112, 113]

Secondly, and building on the assertion that there are gaps in what is known of the experience of immigrating, this systematic review recommends synthesis of qualitative research to better identify the intersectional characteristics that aggravate vulnerability to violence, and

the constructs and distributions of power that produce this violence. This should include a more detailed breakdown of the dependencies and points of vulnerability that are built into family migration categories and the relationship with domestic violence and IPV. Furthermore, while this review included state violence, there was very little systematic and quantitative research that could evidence state violence against people in insecure migration status. While we know that this violence happens from small qualitative studies and case studies, the research that can estimate the scale of this violence is not available. States are notoriously secretive about the violence they perpetrate. The data that was identified in this review related to the United States which is well known for adopting violent policing and immigration and detention tactics. More research on state violence against people in insecure migration status is needed, particularly outside of North America. Additionally, the regional distribution of this review demonstrates clearly that there is a deficit of studies published on violence against people in insecure migration status that happens outside of Europe and North America. While this reflects the Western bias within academic research more generally, it is still more pronounced when placed in the context of migration studies, because far more migration happens outside of Europe and North America than within and towards Europe and North America.

## Conclusion

This review found that physical violence is a widespread issue for people in insecure migration statuses. It found that the prevalence of intimate partner violence against people in insecure status was higher than the rate for the population as whole. It found that people in undocumented statuses did not experience higher prevalence of physical violence than other types of insecure status.

The review suggested that better quantitative data is needed regarding insecure status and associated characteristics, and that the category of 'foreign born' is inadequate to measure phenomena attached to immigration status. The review suggested that qualitative review is needed to elaborate on the intersectional characteristics that may influence experience of violence when in insecure migration status, and in particular to enrich data on gender-based violence. It also suggests that there is a regional bias in available data, and that a multilingual review is necessary to better assess data deficits.

## Supporting information

**S1 Appendix. PROSPERO registration.**
(PDF)

**S2 Appendix. Eligibility criteria.**
(PDF)

**S3 Appendix. Search strategy.**
(PDF)

**S4 Appendix. Insecure status definition.**
(PDF)

**S5 Appendix. Risk of bias table.**
(PDF)

## Author Contributions

**Conceptualization:** Alexandria Innes, Hannah Manzur, Elizabeth Cook, Jessica Corsi.

**Data curation:** Alexandria Innes, Sophie Carlisle, Hannah Manzur.

**Formal analysis:** Sophie Carlisle.

**Investigation:** Sophie Carlisle.

**Methodology:** Alexandria Innes.

**Project administration:** Alexandria Innes.

**Supervision:** Natalia V. Lewis.

**Writing – original draft:** Alexandria Innes, Sophie Carlisle, Elizabeth Cook, Jessica Corsi.

**Writing – review & editing:** Alexandria Innes, Sophie Carlisle, Elizabeth Cook, Jessica Corsi, Natalia V. Lewis.

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
