## [Decision Letter · Decision Letter 0]

9 Nov 2023

PONE-D-23-31719Prevalence of physical violence against people in insecure migration status: A systematic review and meta-analysisPLOS ONE

Dear Dr. Innes,

Thank you for submitting your manuscript to PLOS ONE. After careful consideration, we feel that it has merit but does not fully meet PLOS ONE’s publication criteria as it currently stands. Therefore, we invite you to submit a revised version of the manuscript that addresses the points raised during the review process. Please submit your revised manuscript by Dec 24 2023 11:59PM. If you will need more time than this to complete your revisions, please reply to this message or contact the journal office at plosone@plos.org. Please include the following items when submitting your revised manuscript:A rebuttal letter that responds to each point raised by the academic editor and reviewer(s). You should upload this letter as a separate file labeled 'Response to Reviewers'.A marked-up copy of your manuscript that highlights changes made to the original version. You should upload this as a separate file labeled 'Revised Manuscript with Track Changes'.An unmarked version of your revised paper without tracked changes. You should upload this as a separate file labeled 'Manuscript'.

We look forward to receiving your revised manuscript.

Kind regards,

Cesar Infante Xibille, Ph.D

Academic Editor

PLOS ONE

Journal Requirements:

2. Thank you for stating the following financial disclosure: "All authors (AI, SC, HM, EC, JC, NL) were supported by the the UK Prevention Research Partnership (Violence, Health and Society; MR-VO49879/1)".

3. Please upload a copy of Supporting Information Figure/Table/etc. S1 Fig 1 to S7 Fig 7,  which you refer to in your text on page 36.

Reviewers' comments:

Reviewer's Responses to Questions

**Comments to the Author**

1. Is the manuscript technically sound, and do the data support the conclusions?

Reviewer #1: Yes

Reviewer #2: Partly

Reviewer #3: No

Reviewer #4: Yes

2. Has the statistical analysis been performed appropriately and rigorously? 

Reviewer #1: Yes

Reviewer #2: Yes

Reviewer #3: No

Reviewer #4: Yes

3. Have the authors made all data underlying the findings in their manuscript fully available?

Reviewer #1: Yes

Reviewer #2: Yes

Reviewer #3: No

Reviewer #4: Yes

4. Is the manuscript presented in an intelligible fashion and written in standard English?

Reviewer #1: Yes

Reviewer #2: No

Reviewer #3: Yes

Reviewer #4: Yes

5. Review Comments to the Author

Reviewer #1: The time span and number of migrants of this review is remarkable. Well written. Scientifically sound.

Line 62-71: consider citing references for these comments.

Line 87: Is there a category that accounts for pre-migratory violence? This could be a risk factor for mental health outcomes. Consider saying something about this even though it is not the population of interest in this study.

Line 185: The exclusion of other forms of violence is well explained in the context of scientific rigor.

The methods and analytical tools appear to be quite thorough.

Line 334: refers to eleven studies, then the same twelve studies. Is this an error?

Line 407: I suggest stating the prevalence rate for IPV in insecure migration here so reader can easily compare to the gen pop 23.1%.

Line 538: on what basis do you assert that "particularly liberal states" are notoriously secretive? One might argue that no state is open about the violence they perpetrate. Perhaps there is a debate to be had but it strays from the rest of the discussion, so I suggest removing "and particularly liberal states".

The intro and discussion are true to the study but lacks some engagement of the reader on the clinical implications. Why is it important to study violence? What effect does physical violence have on migrants? No where in the paper is there mention of trauma, mental health or physical health outcomes. You may wish to contract some of the research and policy implications to be more concise and add some empathic, clinical implications in order to contextualize the research and engage readership.

Figures all very well done and give a good snapshot of the data

An important piece of research that contributes new data to the field.

Reviewer #2: Dear Repectable Authors

Thank you for considering a significant area of research related to the health of migration. You investigated the physical violence against people in insecure migration status. Your results is of interest but the way you report the methods and results needs modification.

- Abstract, methods, please add name of databases. Also, add your checklist for quality appraisal and the methods used for meta-analysis. Add the software and replace the PROSPERO information to the end of the abstract.

- Abstract, results, please add the p-values for all outcomes.

- Line 87, please remove this heading.

- Line 116-118 are redundant. Please remove from here.

- LIne 122, eligibility criteria, this section is too long. Please briefly mention only the inclusion and exclusion criteria. Based on the PRISMA (Items 10a and 10b), the rest of the sentences that contain definitions should be mentioned in the data extraction section.

- Line 186, type of study, there is no such a subheading in the PRISMA checklist. This information should be mentioned in the eligibility criteria section.

- Line 189, what is your reason for selecting studies after 2000?

- Lines 189-90 is related to search. Please remove these information from this section.

- What is the reason for not searching major databases such as PubMed and Scopus? I think you missed some studies.

- Line 195, the search period and the exact time of the search must be mentioned. Also, In my opinion, it is better to merge information sources and search strategy under one heading.

- In my opinion, Tables 1 and 2 can be added to the article as supplementary files.

- Based on PRISMA, please add search strategy for all databases as a supplementary file. This file also include the number of records and time of search.

- Line 248, please add the version.

- What about publication bias? Please add funnel plot for at least one outcome.

- Lines 267-272 needs subheading: "study selection", items 16a and 16b of PRISMA checklist.

- Since different tools/questionnaires are used in different studies and the variety of tools can affect the results, I think it is better to add a column to Table 3 and mention the type of tool. Secondly, it is better to add an analysis according to the type of tool to the results.

- LIne 307, please add some percent in this section for your results.

- Please remove subheadings from the discussion section.

- Please add a conclusion.

- Line 51, it is not clear enough. How you interpreted that " the prevalence of violence against people in insecure migration status is high". I search the whole manuscript and there is not a category for your statement or not a reference for it. Please clear it. I agree with you that this is a high value, but it is better to have a reference for this statement.

Cheers

Reviewer #3: If published, the manuscript would be valuable. However, the conclusion of higher/lower prevalence when comparing different sub-groups of study participants was not statistically supported (the compared sub-groups have overlapping confidence intervals). More importantly, the analysis has computational flaws that result in conflicting results (e.g., various prevalence reports for the same study).

Reviewer #4: The authors did a great job of summarizing the evidence of physical violence in people with insecure immigration status. This work is critical to recognize and address the burden of violence in this marginalized population worldwide. Despite several strengths of this work, some issues need further clarity and edits to better communicate the purpose, method, and findings of this meta-analytic study.

First, the authors should consider providing specific examples and relevant references in lines 64-71. Although those points are reasonable, their credibility should be strengthened further with evidence from earlier research.

Second, the authors may wish to add reference(s) for four broader categories they used in lines 88-89 and afterward. Also, they must mention if those categories are relevant to physical violence only or all forms of violence. If later, they should explain how those categories are useful to distinguish physical violence from others and how those constructs are used in this meta-analytic review.

Third, in the background, the authors should strengthen the rationale for this review. A few ideas, such as the current trends in insecure migration globally, their predictors and consequences, and relevance to estimating physical violence, should be discussed, establishing sufficient reasons why this study is needed, or in other words, how this study can help policymakers and practitioners. Those areas should be explored and comprehensively discussed before presenting the aim/purpose of this study. Also, the authors should explain why physical violence is studied in this article rather than psychological, social, or other forms of violence. Any case or type of violence is unacceptable. However, we need to justify why we consider or prioritize one type as a focus of this work.

Fourth, I admire the efforts of the authors to clarify the lack of definition for insecure immigration status and their proposed constructs that may capture this idea in different population groups who don’t have secure immigration status. However, those categories may not be consistent across contexts. For example, employment-based immigrants vs. undocumented immigrants have notable differences in terms of psychosocial, legal, administrative, and other forms of stressors that may affect their vulnerability to violence. Also, specific categories, such as asylum seekers, have different sets of rights and protections in different jurisdictions that may affect their inclusion in insecure status, as mentioned in this study. Their insecurity is generally higher in places of origin rather than where they stayed after voluntary displacement/migration. Those concepts should be clarified either in Table 1 in a separate column and/or within the methods section. Also, having a broader definition of “secure immigration status” with sufficient sources could be helpful.

Fifth, while categorizing the perpetrators, the authors may consider other possible subgroups, such as employers, specific non-state entities, and interpersonal violence that don’t fall in IPV or state-induced violence.

Sixth, please provide a detailed explanation of the risk of bias assessment and how studies with varying risks of bias offered similar/different estimates for physical violence.

Seventh, a funnel plot is necessary to establish whether publication bias was present in this meta-analysis. Please consider using this approach, as publication bias can profoundly impact the credibility and scope of the evidence in meta-research.

Eighth, the role of covariates should be examined using a meta-regression approach. Please explain why that was not conducted in this study, and consider adding meta-regression for key variables/covariates of interest, which may enrich this study.

Lastly, the discussion sufficiently reflects on the prevalence of physical violence but provides limited insights into why the estimates vary across specific subgroups. Also, the social determinants of violence and their roles in burden estimates should be briefly discussed. Possible causes of varying estimates across studies and groups should be explained using previous research so that the readers can contextualize the findings of this meta-analytic study. Furthermore, a brief conclusion with key take-away messages of this study would be helpful.

6. PLOS authors have the option to publish the peer review history of their article (what does this mean?). If published, this will include your full peer review and any attached files.

Reviewer #1: No

Reviewer #2: **Yes: **Morteza Arab-Zozani

Reviewer #3: No

Reviewer #4: No

---

## [Author Response · Author response to Decision Letter 0]

20 Dec 2023

Please find our detailed reponse letter attached, specifying changes to each suggestion. We thank you for your time and for your close engagement with our work.

---

## [Decision Letter · Decision Letter 1]

19 Jan 2024

PONE-D-23-31719R1Prevalence of physical violence against people in insecure migration status: A systematic review and meta-analysisPLOS ONE

Dear Dr. Innes,

Thank you for submitting your manuscript to PLOS ONE. After careful consideration, we feel that it has merit but does not fully meet PLOS ONE’s publication criteria as it currently stands. Therefore, we invite you to submit a revised version of the manuscript that addresses the points raised during the review process.

We look forward to receiving your revised manuscript.

Kind regards,

Cesar Infante Xibille, Ph.D

Academic Editor

PLOS ONE

Journal Requirements:

Reviewers' comments:

Reviewer's Responses to Questions

**Comments to the Author**

1. If the authors have adequately addressed your comments raised in a previous round of review and you feel that this manuscript is now acceptable for publication, you may indicate that here to bypass the “Comments to the Author” section, enter your conflict of interest statement in the “Confidential to Editor” section, and submit your "Accept" recommendation.

Reviewer #2: All comments have been addressed

Reviewer #3: All comments have been addressed

2. Is the manuscript technically sound, and do the data support the conclusions?

Reviewer #2: Yes

Reviewer #3: Yes

3. Has the statistical analysis been performed appropriately and rigorously? 

Reviewer #2: Yes

Reviewer #3: Yes

4. Have the authors made all data underlying the findings in their manuscript fully available?

Reviewer #2: Yes

Reviewer #3: Yes

5. Is the manuscript presented in an intelligible fashion and written in standard English?

Reviewer #2: Yes

Reviewer #3: Yes

6. Review Comments to the Author

Reviewer #2: Dear Respected Authors

Thank you for your detailed answers and helpful corrections.In my opinion and probably the authors themselves, the changes made have increased the quality of your manuscript and made it easier for readers to read it. Your manuscript is acceptable in this fashion.

Cheers

Reviewer #3: We employ two statistical methods to conclude a comparison of two or more statistics, as one is different from (greater or lower than) the other. One is a confidence interval. When the confidence interval excludes the null value or when the two confidence intervals do not overlap, we can conclude that the observed values differ (one is higher or lower than the other). A p-value can also be used to test for statistical significance. A p-value shows the probability of obtaining the observed difference if the null hypothesis of no difference is true.

For example, we may use the p-value of the selected test statistic to assess whether the difference in prevalence for men and women of 7.52% (i.e., 35.30%-27.78%) is attributable to chance alone (random error) or to an actual difference in prevalence among the underlying male and female populations. Similarly, we can create a null hypothesis for the difference (prevalence in males versus prevalence in women) and use a confidence interval to determine whether the observed difference reflects the underlying population difference or not. We cannot conclude that there is a difference between the two populations if the confidence interval contains the stated null value.

Alternatively, for the comparison groups to be different, their confidence intervals must not overlap. In this study, for example, the 95% confidence intervals of 18.45-52.15 for men and 21.42-34.15 for women overlap, rendering the conclusion that men had a larger prevalence than women incorrect. How come men's lower limit of 18.45% prevalence is greater than women's lower limit of 21.42% prevalence?

Fundamentally, both the confidence interval approach and the p-value approach yield the same conclusion for the same data, but this is not maintained in this manuscript under consideration. Except for the absence of clarity of the aforementioned scientific integrity concern, the manuscript is really well written, and I feel it will be very useful in establishing an international intervention approach.

7. PLOS authors have the option to publish the peer review history of their article (what does this mean?). If published, this will include your full peer review and any attached files.

Reviewer #2: **Yes: **Morteza Arab-Zozani

Reviewer #3: No

---

## [Author Response · Author response to Decision Letter 1]

30 Jan 2024

Dear Editors and Anonymous Reviewers,

Thank you again for considering our manuscript ‘Prevalence of Physical Violence Against People in Insecure Migration Status: A Systematic Review’ for publication in PLOS One. We are very grateful for the reviewer comments, and believe that they have supported the development of this manuscript. We were very pleased to have a positive response from Reviewer 2 who considered the manuscript acceptable for publication. 

Reviewer 3 advanced the following query:

We employ two statistical methods to conclude a comparison of two or more statistics, as one is different from (greater or lower than) the other. One is a confidence interval. When the confidence interval excludes the null value or when the two confidence intervals do not overlap, we can conclude that the observed values differ (one is higher or lower than the other). A p-value can also be used to test for statistical significance. A p-value shows the probability of obtaining the observed difference if the null hypothesis of no difference is true.

For example, we may use the p-value of the selected test statistic to assess whether the difference in prevalence for men and women of 7.52% (i.e., 35.30%-27.78%) is attributable to chance alone (random error) or to an actual difference in prevalence among the underlying male and female populations. Similarly, we can create a null hypothesis for the difference (prevalence in males versus prevalence in women) and use a confidence interval to determine whether the observed difference reflects the underlying population difference or not. We cannot conclude that there is a difference between the two populations if the confidence interval contains the stated null value.

Alternatively, for the comparison groups to be different, their confidence intervals must not overlap. In this study, for example, the 95% confidence intervals of 18.45-52.15 for men and 21.42-34.15 for women overlap, rendering the conclusion that men had a larger prevalence than women incorrect. How come men's lower limit of 18.45% prevalence is greater than women's lower limit of 21.42% prevalence?

Fundamentally, both the confidence interval approach and the p-value approach yield the same conclusion for the same data, but this is not maintained in this manuscript under consideration. Except for the absence of clarity of the aforementioned scientific integrity concern, the manuscript is really well written, and I feel it will be very useful in establishing an international intervention approach.

Thank you for the detailed comment. Reviewer 2 asked to add the p-values for all outcomes throughout abstract and results which we did. In response to your comment, we have reconsidered our reporting and adjusted the language to ensure that we are very clear that there is no statistically significant difference between the prevalence estimates.

You will find the changes to reflect this tracked at lines: 41-52, 340-341, 359-366, 372-378, 391-392, 397-399 and 433-434.

We hope that these changes satisfy the concerns of Reviewer 3. Thank you again for your time reviewing this work. Please do contact me at alexandria.innes@city.ac.uk if there is anything further we can provide.

We will look forward to hearing from you in due course.

Sincerely yours,

Alexandria Innes

---

## [Decision Letter · Decision Letter 2]

23 Feb 2024

Prevalence of physical violence against people in insecure migration status: A systematic review and meta-analysis

PONE-D-23-31719R2

Dear Alexandria Innes

We’re pleased to inform you that your manuscript has been judged scientifically suitable for publication and will be formally accepted for publication once it meets all outstanding technical requirements.

Kind regards,

Cesar Infante Xibille, Ph.D

Academic Editor

PLOS ONE

Additional Editor Comments (optional):

Reviewers' comments:

Reviewer's Responses to Questions

**Comments to the Author**

1. If the authors have adequately addressed your comments raised in a previous round of review and you feel that this manuscript is now acceptable for publication, you may indicate that here to bypass the “Comments to the Author” section, enter your conflict of interest statement in the “Confidential to Editor” section, and submit your "Accept" recommendation.

Reviewer #3: (No Response)

2. Is the manuscript technically sound, and do the data support the conclusions?

Reviewer #3: Yes

3. Has the statistical analysis been performed appropriately and rigorously? 

Reviewer #3: Yes

4. Have the authors made all data underlying the findings in their manuscript fully available?

Reviewer #3: Yes

5. Is the manuscript presented in an intelligible fashion and written in standard English?

Reviewer #3: Yes

6. Review Comments to the Author

Reviewer #3: Dear Authors,

You did a wonderful job and I recommend the acceptance of your manuscript for publication. However, exclude the p-values or provide properly determined p-values (that is compatible with the provided confidence intervals) for differences in two proportions. A confidence interval and p-value must result in the same conclusion for the same data. In this paper, the confidence intervals overlap, indicating that there is no statistically significant difference between the two proportions, such as males versus women, but not limited to, whereas the p-value is less than 0.05 (p<0.00 for all the comparisons), indicating a significant difference. I suggest you engage with a statistician to resolve this issue.

Kindest regards,

7. PLOS authors have the option to publish the peer review history of their article (what does this mean?). If published, this will include your full peer review and any attached files.

Reviewer #3: No

---

## [Editor Report · Acceptance letter]

6 Mar 2024

PONE-D-23-31719R2 

PLOS ONE

Dear Dr. Innes, 

I'm pleased to inform you that your manuscript has been deemed suitable for publication in PLOS ONE. Congratulations! Your manuscript is now being handed over to our production team.

Kind regards, 

on behalf of

Dr. Cesar Infante Xibille 

Academic Editor

PLOS ONE